# Cross-Border Mergers and Acquisitions as a Challenge for Sustainable Business

**Jaroslava Hečková [1], Róbert Štefko [2], Miroslav Frankovský [3], Zuzana Birknerová [3], Alexandra Chapčáková [1] and Lucia Zbihlejová [4,*]**

[1] Department of Economics and Economy, Faculty of Management, University of Prešov in Prešov, Konštantínova 16, 080 01 Prešov, Slovakia; jaroslava.heckova@unipo.sk (J.H.); alexandra.chapcakova@unipo.sk (A.C.)

[2] Department of Marketing and International Trade, Faculty of Management, University of Prešov in Prešov, Konštantínova 16, 080 01 Prešov, Slovakia; robert.stefko@unipo.sk

[3] Department of Managerial Psychology, Faculty of Management, University of Prešov in Prešov, Konštantínova 16, 080 01 Prešov, Slovakia; miroslav.frankovsky@unipo.sk (M.F.); zuzana.birknerova@unipo.sk (Z.B.)

[4] Department of Intercultural Communication, Faculty of Management, University of Prešov in Prešov, Konštantínova 16, 080 01 Prešov, Slovakia

\* Correspondence: lucia.zbihlejova@unipo.sk

**Abstract:** When considering the challenges for sustainable business, companies implementing cross-border reallocation of capital by means of mergers and acquisitions should take into account the context and evaluation of attributes of their future implementation. The main aim of the paper is, therefore, to identify and specify the key attributes of sustainable cross-border mergers or acquisitions (M&As) influencing the considerations about their future implementation. On the basis of the views of managers from 120 companies (international corporations selected from the Zephyr database) located in 45 countries within the European area that had previously been the subject of a cross-border merger or acquisition, significant attributes were extracted in connection with their experience from their implementation. These attributes are taken into account when considering the implementation of a cross-border merger or acquisition in the future. A factor analysis of the data obtained allowed the extraction of three key attributes of implementation of a potential merger and acquisition process as an important tool of business sustainability—aims, concerns, and reasons. This paper further presents the basic parameters of the Attributes of Future Mergers and Acquisitions (AFM&A) methodology: eigenvalues, Cronbach's alpha values, the percentage of the variance explained, inter-correlations of the extracted factors, and the results of an analysis of differences in the assessment of the extracted factors by managers. At the same time, no statistically significant differences were found in the assessment of the extracted merger and acquisition assessment factors. The study fills in the research gap in the area by identifying and specifying the attributes of considering the future implementation of M&A management in terms of the broader concept of this issue.

**Keywords:** cross-border mergers; cross-border acquisitions; sustainable business; future cross-border M&A activity; future M&A management

## 1. Introduction

Capital reallocation through implementation of cross-border mergers and acquisitions (M&As) represents one of the most significant phenomena of the last two decades in the global as well as the European scale. Global M&A deal volume reached $3.0 trillion in the first 3Q2018, representing 27.0% increase compared to the same period of the previous year [1]. The rankings data includes mergers,

acquisitions, joint ventures, spin-offs, debt-for-equity swaps, private placements of common equity and convertible securities, divestitures, and the cash injection component of recapitalization in accordance with the Bloomberg standards [1].

If M&As are performed well, they can serve as one of the most powerful tools and a strong basis for the corporate growth and survival of a company. Through M&As, firms can be merged and their businesses reconfigured, which means that their assets and resources can be added, redeployed, divested or recombined in order to strengthen the resource base [2]. If M&As are not performed well, they can result in decline or failure, which causes a decrease in the firm's value. According to several studies devoted to this issue, 70% of M&A deals fail to reach their goals [3,4].

Mergers and acquisitions are individual cases of voluntary concentration of two or more independent enterprises into a single entity or, in other words, acquiring ownership and managerial control of one enterprise over another while simultaneously acquiring a controlling interest in the enterprise's voting rights. This is a complex transaction outside the normal business management of the company over a long period of time, fulfilled by an agreement based on relevant financial and factual information.

Within the European area, financial and trade liberalization in the European Union and the European Monetary Union has made a significant contribution to this. Cross-border mergers and acquisitions thus represent a significant global phenomenon that enables businesses to create synergetic effects, acquire discounted assets, create tax savings, gain access to new technologies, expand, differentiate, and diversify business activities, thereby increasing competitiveness and market value, and promoting and possibly securing the business sustainability [5,6].

Implementation and efficiency of the merger and acquisition processes are multifactorially conditioned. In these processes, factors of a different macro- and microeconomic nature are introduced (partial conclusions in [7–12]). One of the important aspects is their subjective perception. Within this concept, the presented contribution focuses on the managerial view and identification of the attributes of the considered implementation of a cross-border merger or acquisition in the future.

The main aim of this paper is to identify and specify the key attributes of sustainable cross-border mergers or acquisitions that influence the considerations about their future implementation. Our motivation for implementation of this research was the effort to shift the knowledge in the subject area of research, as well as the absence of scientific studies dealing with the area of potential intent to implement a cross-border merger or acquisition (from the point of view of the companies themselves), as most M&A scientific studies address only the aspects relating to the before-during-and-post-merger and acquisition processes (see the Literature Review).

Another of our motivations came precisely from the concept of behavioral economics, which we applied in our research, and which supports, by synergetic effects, explication of the potential of economic sciences in terms of more realistic psychological aspects of economic and managerial actions. The concept of behavioral economics is currently one of the fascinating fields of integration of psychological phenomena into economic models, so that they predict more accurately and reliably human behavior and decision-making. At the heart of our interest is to explore the attributes of a potential M&A implementation in the context of sustainability principles, which are beginning to integrate into the business practice.

According to the Brundtland Report issued by the Brundtland Commission as the World Commission for Environment and Development (WCED), a UN Advisory Body [13], business sustainability can be defined as the development of the needs and requirements of the present, without compromising the ability of future generations to meet their own needs; this aspect has recently become the subject of interest for businesses.

The essence of business sustainability is focusing on a holistic concept of assessing overall business performance through the integration of environmental, social, and economic sustainability [14]. According to [15], significant emphasis is currently placed on improving the quality of corporate

social performance, including the incorporation of environmental, social, and governance factors into business practices.

To develop this issue further, the paper is therefore structured as follows: the second section provides a brief M&A literature review, focusing on the business sustainability as linked with cross-border mergers and acquisitions, including the area of our interest and research. It is the area of specification of key attributes of the implementation of the future merger and acquisition process. Subsequently, we formulate hypotheses regarding the assumption of the existence of the concept of the multifactor structure of the assessment of merger and acquisition processes, as well as the existence of statistically significant differences in the assessment of the extracted factors of the mentioned multifactor structure of M&A processes. The third section of the article describes our own research methodology for the detection of the key attributes of implementation of future cross-border mergers and acquisitions, including the descriptions of data collection and the research sample, as well as the methodological apparatus (analytical methods). In the follow-up section, we describe the results of our research and in the final section we present the discussion and conclusions on the results achieved, including the implications for further research in the subject area of research.

## 2. Literature Review and Hypotheses

Mergers and acquisitions are a common designation for transactions relating to the purchase and sale of businesses, parts of businesses, assets, shares, and business units. The creation of capital-linked national or international businesses brings benefits not only to the participating subjects, but also, to a certain extent, to the whole company and its further sustainability [6]. Defining the boundaries where large corporations that have gained power through mergers and acquisitions contribute to creating prosperity for the whole company, and where they already contribute to the destruction of the positive development (cartels, abuse of monopolies, loss of scale) is an issue which has not yet been definitely solved [7,16–18].

In the relevant scientific literature, most publications and studies present the enormous importance of M&A transactions in the context of growth and profit targets and the accompanying value increase of an enterprise [5], but they lack detailed information on the process of structuring the M&A transactions in the context of business sustainability.

We agree with the opinions of [6] that, not surprisingly, for the past decades, a number of researchers have dealt with different factors—primarily, according to our opinion, the factors relating to the before-during-and-after merger and acquisition process [19–34], including our own research so far [12,35,36] affecting the sustainability of M&As.

More recently, researchers have been focusing on the human factors and sociocultural variables which contribute to the sustainability of M&A process in the context of examining what impact trust has on a successful and sustainable integration of two firms in terms of the viewpoints of both the acquiring and the acquired companies' management [6], then, in the context of the factors which are crucial for sustainable M&A transactions in companies, in terms of the holistic concept, which has been developed particularly for the national and international M&A processes in order to test the service market as well as for the service sector market in general [5]. This is also true in the context of studying the sociocultural links, such as cultural integration, trust, complementary employee skills, or collective teaching, between the merging firms, which have an impact on the level of knowledge transfer in M&As, conditioned particularly by the human resources flexibility (flexibility in employee skills, flexibility in employee behavior, and flexibility in HR practices) [37]. A crucial yet underexplored field is the area of potential considerations of implementation of a merger or acquisition which represents the research subject of this scientific study and, at the same time, presents a shift in knowledge in this area. Based on our extensive review of the published literature in this field, no similar publication outputs were found.

Three hypotheses were formulated for the proposed research:

**H1:** *Existence of the concept of a multifactor structure of assessment of merger and acquisition processes, which, in the context of behavioral economics, allows a more comprehensive view of this issue, is assumed.*

**H2:** *Existence of statistically significant differences in the assessment of the extracted factors of the above-mentioned multifactor structure of merger and acquisition processes assessment is assumed.*

**H3:** *Existence of statistically significant differences in the assessment of the selected factors of the above-mentioned multifactor structure of merger and acquisition processes assessment between the two sectors examined is assumed.*

## 3. Materials and Methods

Identifying the attributes of the considered implementation of a cross-border merger or acquisitions in the future was carried out using the questionnaire method. The main objective of the research was to identify and specify the factors which relate to the considerations about a merger or an acquisition process in the future.

Based on the theoretical elaboration of the issue, the objective of the conducted research was to verify the factor structure of the original methodology AFM&A – Attributes of Future Mergers and Acquisitions (the questionnaire items are shown in Table 1). The selection and formulation of the items in the methodologies were made on the basis of the findings from the previous research studies [7–12]. The task for the managers was to assess the individual items of the questionnaire on a 4-point scale, where 1 = insignificant, and 4 = very significant (see Appendix A). In the first quarter of 2017, 1,000 business companies were contacted to create a research sample based on a mail communication with an online questionnaire link.

**Table 1.** AFM&A methodology factor structure.

| Questionnaire Items | Factors | | |
|---|---|---|---|
| | 1—Aims | 2—Concerns | 3—Reasons |
| Lack of control over acquisition | 0.064 | **0.851** | 0.187 |
| Transition management | 0.046 | **0.541** | -0.195 |
| Staffing issues | 0.063 | −0.362 | **0.745** |
| Adding clients | 0.232 | **0.749** | −0.027 |
| Expand geographically | −0.114 | 0.195 | **0.704** |
| Adding staff | −0.094 | −0.036 | **0.780** |
| Expand firm's financial resources | **0.886** | 0.134 | −0.147 |
| Manage growth | **0.806** | 0.111 | −0.123 |
| Improve firm's management resources | **0.894** | 0.175 | −0.124 |
| Provide impetus for growth | **0.905** | 0.272 | −0.037 |
| Increase opportunities for staff | **0.844** | −0.161 | 0.161 |

Businesses were selected from the Zephyr database [38]. These were companies that had been subject to cross-border mergers and acquisitions in the period of 2010–2016. Thirty days after the questionnaire was sent, the CEOs or top managers of these companies were sent a reminder mail. Out of the total of 1,000 business companies approached, 120 business companies completed the questionnaire. Respecting the definition of an enterprise of the European Commission Recommendation no. 2003/361/EC (Annex 1, Article 2: size defined according to the number of employees and the annual balance sheet total and/or annual turnover) [39], the size of these companies was in the categories of small, medium, and large enterprises with market capitalization (total value of the company's shares in circulation, calculated by multiplying the number of shares issued by the company and the price of those shares on the market) of more than €100 mil. in 45 countries of European Economic Area (Austria, Albania, Armenia, Azerbaijan (European part), Belarus, Belgium, Bosnia and Herzegovina, Bulgaria, Croatia, the Republic of Cyprus, Czech Republic, Denmark, Estonia, Finland, France, Greece, Georgia (European part), Hungary, Ireland, Kazakhstan (European part), Netherlands, Norway, Latvia, Lithuania, Liechtenstein, Luxembourg, Hungary, the Republic of Malta, the Federal Republic of Germany, Poland, Portugal, Romania, Slovakia, Slovenia, Spain, Sweden, Switzerland, Italy, Turkey (European part),

United Kingdom, Russia (European part), Ukraine, Serbia, Moldova, Iceland). This selection may be considered as deliberate and, at the same time, it is of a voluntary nature. The method of selecting companies was also related to the reasonable degree of generalization of the results obtained.

The final research sample thus consisted of 108 male and 12 female managers aged between 21 and 65 years (M = 42.90 years, SD = 11.270 years), who had been working for their company from 1 to 25 years (M = 11.50 years, SD = 6.118 years). These managers were working on a position in their company's top management, with 75 (63%) managers working in the manufacturing sector and 45 (37%) managers working in the sector of services.

The data obtained were evaluated by means of the mathematical and statistical methods of descriptive statistics, Principal Component Analysis (PCA) with Varimax Rotation, the Pearson Correlation Coefficient, and the Friedman test.

## 4. Results

The factor analysis of the data obtained using the Principal Component Analysis with Varimax Rotation method allowed the extraction of three key attributes of implementation of a potential merger and acquisition process – aims, concerns, and reasons (Figure 1, Tables 1 and 2).

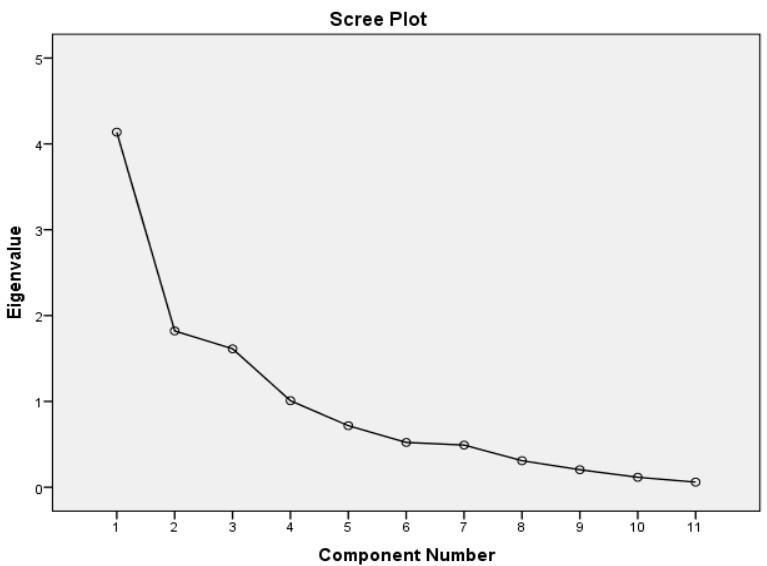

**Figure 1.** Extracted factors of the AFM&A methodology.

**Table 2.** Eigenvalues and the percentage of the variance explained for the extracted factors.

| Factors | Eigenvalues | % of Variance | Cumulative % |
|---------|-------------|---------------|--------------|
| Aims | 4.137 | 37.610 | 37.610 |
| Concerns | 1.821 | 16.555 | 54.165 |
| Reasons | 1.613 | 14.664 | 68.829 |

The extracted factors related to considerations about a future merger or acquisition explain 68.8% of the variance, and it is possible to specify them content-wise as follows:

Factor 1—**Aims** ($\alpha = 0.925$). When considering a possible merger or acquisition, managers scoring high in this attribute attach greater importance to the expansion of the firm's financial resources, growth management, improvement of the firm's management resources, providing an impetus for growth, and increasing opportunities for the staff. Contents of the items saturating the given factor are specified below.

*Expand firm's financial resources*: Besides the strategic motives, such as geographic diversification, restructuring of entities, widening the range offered, market share increase, or transfer of know-how,

there are also certain financial motives behind all mergers and acquisitions, return of the investment being the most prominent one. A merger or acquisition may result in a larger and a financially more stable enterprise with improved capital structure, and better access to lower-cost credit facilities (lower cost of foreign capital; [40]).

*Manage growth*: Growth of the company in the future is one of the primary motives for implementation of mergers and acquisitions. Businesses trying to grow must choose between an internal, that is, organic growth and a growth through mergers and acquisitions. The internal growth usually represents a slow and uncertain process, whereas the growth through mergers and acquisitions is a much faster process, although it brings along many risks. According to [41], mergers and acquisitions are one of the most important events in corporate finance for firms as well as the economy [42] in order for the modern companies to survive and grow in competitive markets.

*Improve firm´s management resources*: The aim of mergers or acquisitions can also be an attempt to acquire a specific skill (in the case of personal questions) or a resource owned by another business. This type of connection occurs mainly when the smaller enterprise has developed specific high value added skills over several years, and the acquiring enterprise would need a long time to develop the same set of skills, requiring significant investment. Cross-border mergers also make it possible for the companies to acquire resources, such as knowledge base, technologies, and human resources [43], from local firms, as well as gain access to markets and to key constituencies at the local level.

*Provide impetus for growth*: Without a doubt, it is possible to claim that the primary goals of a company and its management include growth. This growth should be accompanied by generating reasonable returns for its owners. Mostly, the management of the company can continue to generate acceptable returns for the owners to maintain a business of a certain size, but the impetus for "aggressive" growth is primarily the decision to reallocate the company's capital through mergers or acquisitions.

Factor 2—**Concerns** ($\alpha = 0.801$). When considering a possible merger or acquisition, managers scoring high in this attribute attach greater importance to the lack of control over acquisition, transition management, and adding clients. Contents of the items saturating the given factor are specified below.

*Lack of control over acquisition:* Control is an important part of the management process and is in fact overlapping with all of its activities. Management of the merger or acquisition process requires thorough control of all partial processes. The shortage or loss of control over this process is the path to the failure of the entire transaction.

*Transition management:* The contextual attribute, which companies take into account when considering implementation of a merger or an acquisition, is one of the issues of change management. Change management addresses aspects that relate to new areas of an "expanded entrepreneurship" in order to achieve the company's strategic goals following a merger while enhancing managerial efficiency and performance (more in [44]).

*Adding clients:* Mergers and acquisitions can also be driven by an attempt to gain access to new markets and a related new client base. For instance, in a case of one bank merging with another one, each of the merged banks acquires the client base of the other bank. There are also cases when the acquired client base represents a market which was previously unavailable.

Factor 3—**Reasons** ($\alpha = 0.815$). When considering a possible merger or acquisition, managers scoring high in this attribute attach greater importance to staffing issues, geographical expansion, and adding staff. Contents of the items saturating the given factor are specified below.

*Staffing issues*: Personnel issues of creating an appropriate organizational structure must be in line with the specific needs of ensuring a smooth operation of the "expanded" entrepreneurship (after a merger or an acquisition) and its goals. The experience and skills of the workforce should be documented and compared with the current as well as the future requirements of the "new" enterprise.

*Expand geographically:* Geographic expansion means that an enterprise seeks to expand its business activities to markets in other regions. In international expansion, an enterprise needs to know the nuances of the new markets, and it is the mergers and acquisitions, joint ventures, or strategic alliances

which can be a solution in this case. Companies with successful products in one national market can, through a cross-border merger or acquisition, achieve revenue and profit growth [45]).

*Adding staff:* In the light of the above characteristics, a cross-border agreement may allow the acquirer to use the domestic labor force. It is about gaining new employees for both the managerial and the non-managerial positions.

The calculated inter-correlation coefficient values of the individual extracted factors (Table 3) support the proposed AFM&A methodology factor structure.

**Table 3.** Inter-correlation coefficient values of the AFM&A factors.

|  |  | **Concerns** | **Reasons** |
|---|---|---|---|
| **Aims** | Pearson Correlation | 0.266 | −0.138 |
|  | Sig. (2-tailed) | 0.003 | 0.132 |
| **Concerns** | Pearson Correlation |  | −0.085 |
|  | Sig. (2-tailed) |  | 0.356 |

The results of the correlation analysis confirmed the statistically significant positive correlation only between the Aims and Concerns attributes. This result supports the fact that, when considering implementation of a future merger or acquisition, the greater the importance the managers attach to the attribute of Aims, the more importance they attach also to the attribute of Concerns. The greater the importance attributed to the expansion of the firm's financial resources, growth management, improvement of the firm's management resources, provision of an impetus for growth, and increasing the opportunities for staff, the greater the importance attributed to the lack of control over acquisition, transition management, and adding clients. No statistically significant correlations were detected between the attributes of Aims and Reasons, and between Concerns and Reasons. The correlation coefficient values point to the fact that the extracted factors identify and specify various attributes related to the considerations of implementation of a future merger or acquisition.

Based on the results of the analyses, it is possible to consider the H1 (assumed existence of the concept of the multifactor structure of the assessment of merger and acquisition processes) as verified and supported.

Comparisons of statistical significance of the differences in the assessment of the extracted factors are illustrated in Table 4.

**Table 4.** Differences in the assessment of the individual factors of the AFM&A methodology.

| **Factors** | **M** | **SD** |
|---|---|---|
| Aims | 2.8133 | 0.72956 |
| Concerns | 2.9111 | 0.52524 |
| Reasons | 2.7667 | 0.63599 |

These comparisons were performed by the Friedman test. The result of this test was not statistically significant (significance level of 0.518). Overall data analysis did not confirm the existence of statistically significant differences in the assessment of the extracted factors. The mean values of the factors' assessment suggest that they were considered significant by the managers.

Based on the results of the analyses, it is possible to regard the H2 (assumed existence of statistically significant differences in the assessment of the extracted factors of the multifactor structure of the assessment of merger and acquisition processes) as not supported. It is possible to accept the null hypothesis that there are no statistically significant differences in the assessment of the extracted factors, which means that the managers of the investigated companies have assessed the factors of the assessment of merger and acquisition processes at the same level as significant.

Comparisons of statistical significance of the differences in the assessment of the selected factors between the manufacturing and the services sector are illustrated in Table 5.

**Table 5.** Differences in the assessment of the individual AFM&A factors according to the sectors.

| Factors | Sector | M | SD |
|---------|--------|---|-----|
| Aims | Manufacturing | 2.9778 | 0.35454 |
| | Services | 2.3500 | 1.08360 |
| Concerns | Manufacturing | 2.9630 | 0.33359 |
| | Services | 2.8333 | 0.84667 |
| Reasons | Manufacturing | 2.9074 | 0.48522 |
| | Services | 2.5000 | 0.86343 |

The factor assessment comparisons within the manufacturing sector did not confirm the existence of statistically significant differences in the assessment of the extracted factors (significance level of 0.347). However, in terms of the services sector, statistically significant differences were recorded in the assessment of the extracted factors (significance level of 0.006), with the highest score achieved in the factor of Concerns, and the lowest score achieved in the factor of Aims. This finding supports the necessity to analyze the data acquired with regard to the sector in which the companies considering a future implementation of a cross-border merger or acquisition operate.

Based on the results of the analyses, it is possible to consider the H3 (assumed existence of the statistically significant differences in the assessment of the selected factors of the above-mentioned multifactor structure of merger and acquisition processes assessment between the two sectors examined) as verified and supported.

## 5. Discussion and Conclusions

The research of the current state of knowledge of the issue of mergers and acquisitions in the economic, legal, and managerial fields, as well as the own empirical research conducted in several stages brought a number of partial results and conclusions (more in [10–12]). Despite the extensive empirical material focusing exclusively on the M&A process and its individual stages as such, which served as the basis for the analysis, it was not possible to evaluate all the specifics and possible combinations of the factors that may influence the considerations about an implementation of a cross-border merger or acquisition in the future, either in a positive or a negative sense. Considering a future merger or acquisition as a possible alternative for achieving the strategic goals of a company takes into account the whole set of facts.

As acquisition requires significant changes in both companies, their integration may be easier if they are flexible enough after the acquisition is implemented [3,10,20,23]. These changes are most prominent in the systems of planning, control, reporting structures, compensation, and IT of the newly created company [3,17,18]. Those companies, which are more experienced in dealing with these changes, are better at securing the transition, which ultimately leads to greater synergy (more in [28,29]). They are also more skilled in negotiations, understand the target selection process better, and are better at running their operations smoothly along with the acquired companies [16,22,25]. As they might expect a certain level of resistance to change from the managers of the acquired company, as well as from their own lower-level managers, they use their experience from the previous change processes to overcome or even prevent this resistance [37]. The experienced organizations are highly capable of selecting the right companies for acquisition and securing their purchase and integration [25,30]. According to [46], organizational learning is also one of the essential prerequisites for implementing an effective acquisition, as the experienced companies learn from the previous large-scale changes and this knowledge can be further applied to another acquisition process, especially after its completion [25].

As opposed to internal ventures, acquisitions are more capable of preventing the company from developing proprietary skills to endow it with a competitive advantage [47,48].

Managers' considerations involve the lack of control over the acquisition, transition management, staffing issues, adding clients, expanding geographically, expanding the firm's financial resources, growth management, improving firm's management resources, providing impetus for growth, and increasing opportunities for staff. Results of the factor analysis, the Cronbach's alpha calculations, and the inter-correlation analysis prove that the most important factors of considerations of a future merger or acquisition are Aims, Concerns, and Reasons. The acquisition strategy may be therefore seen as a path to the growth of the value of the company, or a tool for restructuring the corporate structure and a risk diversification tool. The results obtained confirm the fact that the merger and acquisition business strategy is related not only to finance and investment but also to the exchange of know-how, market position, enterprise value, risk reduction, transaction costs, restructuring, including other factors crucial for sustainable business.

The presented results have enriched the knowledge in the area of mergers and acquisitions both at the methodological and the theoretical level. Methodologically, the knowledge in the area has been expanded by designing and validating the new, original AFM&A methodology, the use of which can contribute to the efficiency increase when considering implementation of a future merger or acquisition. From a theoretical viewpoint, the contribution is enhanced by the identification and the further specification of the basic attributes of the process of considering the implementation of mergers and acquisitions. Extraction of these attributes makes it possible not only to define the theoretical concept of these processes, but also, by implementing them in practice, to contribute to increasing the efficiency of decision-making about the implementation of mergers and acquisitions, and to maintaining the effects of these processes in the long run [12], thus securing business sustainability [5,6].

Identification and specification of the attributes of considerations about a future M&A management implementation in terms of the wider concept of this issue is a concept that enriches knowledge in this area by filling in the research gap, as the research presented so far focuses primarily on the economic analyses of merger and acquisition effects in the context of the before-during-and-after M&A process.

It is worth highlighting the question of the meaningful generalization degree of defining and assessing these factors, or the abstraction degree which is inevitable for this definition. It is also crucial to consider the issue of trans-situationality and versatility of assessment of these factors when considering mergers and acquisitions, for example, within different sectors, cultures, and so on.

*Future Orientation and Study Limitations*

From the viewpoint of the future orientation of research projects, it is essential to focus on the holistic concept of business sustainability in the context of a comprehensive understanding of this issue and the integration of environmental, social, and economic sustainability [14]. In this sense, it is also necessary to draw attention to the influence of sociocultural, global [6], and human factors [37,49–51], as well as consider analyzing the issue in terms of the economic sectors within which the businesses operate.

There is a discussion relating to these ideas about the possible level of generalization of the lessons learned, which can be regarded as one of the important limiting factors of research in this field. Specifically, it is crucial to consider, especially from the point of view of cross-border mergers and acquisitions, to what extent are the acquired knowledge and its use transcultural, or to what extent and how a particular culture can influence decision-making on merger and acquisition processes in the context of business sustainability [5]. Company managers thus face the issue of local behavior and global thinking integration [52]. For instance, a study [53] conducted within the U.S. settings attempted to determine the most influential factors that affect corporate acquisitions in the U.S. lodging industry. Although it was limited to the issue of the pre-acquisition process and the post-acquisition integration, it accentuated yet another limiting factor of this research study, which is the size and heterogeneity of the research sample that does not enable comparisons of the responses of the company managers from

various cultures. Other limiting factors which should be taken into account in future research on this issue include, for example, using the self-report methodologies, the fact that the assessments of the respondents may not be consistent across time, and that the scope of this research study was limited to the selected variables.

**Author Contributions:** Investigation, supervision, and resources, J.H.; Funding acquisition and project administration, R.Š.; Conceptualization, Z.B. and A.C.; Formal analysis and software, M.F.; Methodology, M.F. and Z.B.; Validation, Z.B.; Data curation and visualization, A.C.; Writing – original draft, review & editing, L.Z.

**Funding:** This research was funded by Vedecká Grantová Agentúra MŠVVaŠ SR a SAV (1/0031/17).

**Conflicts of Interest:** The authors declare no conflict of interest. The funders had no role in the design of the study; in the collection, analyses, or interpretation of data; in the writing of the manuscript, or in the decision to publish the results.

## Appendix A

### Research Questionnaire.

Section A: Background Information, Firm Profile

1. Age: …………………………….

2. Gender:     □ male     □ female

3. How long have you worked in the firm? …………………………….

4. In which sector does your firm operate?     □ manufacturing     □ services

5. Which of the following categories does your firm fall into?

| Staff size | | Annual turnover and/or Annual Balance Sheet Total | |
|---|---|---|---|
| 0 – 9 | | less than € 2 million | |
| 10 – 49 | | € 2 – 9.9 million | |
| 50 – 249 | | € 10 – 49.9 million | |
| 250 – 499 | | € 50 million or more | |
| 500 and above | | | |

Section B: Considering a Future M&A Activity

Assess each of the issues connected to a potential implementation of a future cross-border M&A on a scale of 1 to 4 where 1 = insignificant and 4 = very significant:

| Questionnaire Items | 1 = insignificant, and 4 = very significant | | | |
|---|---|---|---|---|
| Lack of control over acquisition | 1 | **2** | 3 | 4 |
| Transition management | 1 | **2** | 3 | 4 |
| Staffing issues | 1 | **2** | 3 | 4 |
| Adding clients | 1 | **2** | 3 | 4 |
| Expand geographically | 1 | **2** | 3 | 4 |
| Adding staff | 1 | **2** | 3 | 4 |
| Expand firm's financial resources | 1 | **2** | 3 | 4 |
| Manage growth | 1 | **2** | 3 | 4 |
| Improve firm's management resources | 1 | **2** | 3 | 4 |
| Provide impetus for growth | 1 | **2** | 3 | 4 |
| Increase opportunities for staff | 1 | **2** | 3 | 4 |

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
