# Peer review of "Cross-Border Mergers and Acquisitions as a Challenge for Sustainable Business"

_sustainability, doi:10.3390/su11113130_

Reviewer 1 Report

I would like to thank the editor for the chance of writing review to the paper entitled: “Cross-Border Mergers and Acquisitions as 3 a Challenge for Sustainable Business” submitted to Sustainability journal.

I found the paper suitable to the journal publishing area, it is interesting and well written however as every scientific elaboration it has some crucial improvements necessity:

Despite the paper title, the aim is not containing the sustainability aspects but it should!

What does it mean “45 countries of the European area” ??? Europe or EU or other area ?

There is no results presentation in the abstract, you have to allure readers to read your paper in the abstract.

After few lines of introduction (in my opinion it must be rewritten to present study motivation!), the Authors started to present methodology. I suggest to present strong literature review and elaborate study hypothesis based on the review.

There is too little information about the study sample (just the age and gender) the authors should develop this description. There is lack of data collection period.

Results are quite interesting and well elaborated but results discussion in relation to the literature is poor. Authors must clearly state what is similar, what is different, what is new in their findings! Not just write “such as [19-25, 11, 15] and others” it is a simplification.

Conclusion is very narrow: no recommendation, no study limitations, no future study directions

I hope my remarks allow authors to improve their paper.

Good luck

Author Response

Dear reviewer,

we have read your suggestions and decided to follow them. The final manuscript thus contains the following:

1) the main aim of the paper contains the sustainability aspect now

2) “45 countries of the European area” - explained

3) the results are already presented in the abstract as well

4) introduction + literature review + hypotheses - revised and enriched

5) research sample described in greater detail

6) discussion and conclusion enriched according to your comments

Thank you very much for great comments and wise suggestions.

Reviewer 2 Report

I thank the Editor for the opportunity to review “Cross-Border Mergers and Acquisitions as a Challenge for Sustainable Business” submitted to the Sustainability. I appreciated the main topic of the paper. However, I think that it not present the minimum standard required for a publication in an international Journal.

In particular,

·         The theme of sustainability, the main topic of the Journal, indicated in the title and in the keywords is treated very marginally.

·         Introduction.  This section is completely inadequate. The structure and the organization are not in line with a research paper. In fact, it does not present at all the context in which the study takes place, the objective of the paper, what previous studies stated about the topic, the research gap, the contribution of this article and the remainder of you article. In general, there is no clear explanation about the main purpose of the paper.

·         There is not a paragraph about literature review that is essential in a research paper in order to support the study. It should present a focused and a carefully structured outline of what others have done in the topic that the Authors analyzed.

·         Material and Methodology. First of all, I suggest renaming the paragraph in “Methodology”. The methodology is not clear. For example, the choice of the period (2010-2016), the number of countries, the level of capitalization are not well explained.

·         There are not hypothesis and/or RQ but it is essential deriving and/or motivating your empirical work and clarifying your idea in readers’ minds.

·         Results, discussion and conclusions: The results do not seem relevant and well explained. Moreover, in the three sections there is not a comparison with previous studies and no implications are well proposed at all (only methodological and theoretical level). Moreover, limitations and suggestions for further research are limited.

I hope my suggestions can be useful for reviewing in the future the work.

Best regards

Author Response

Dear reviewer,

we have read your suggestions and decided to follow them. The final manuscript thus contains the following:

1) in our opinion, cross-border mergers and acquisitions are a significant prerequisite for sustainable business and the title + abstract + keywords treat it accordingly

2) introduction + literature review - revised and enriched

3) Material and Methodology section - renamed. revised and enriched

4) hypotheses added

5) discussion and conclusion enriched according to your comments

Thank you very much for  wise suggestions.

Round  2

Reviewer 1 Report

The paper has been seriously rewritten. I accept it as it is

Author Response

Dear Reviewer, thank you very much for the positive feedback, we appreciate it highly.

Reviewer 2 Report

I thank the Editor for the opportunity to review the new version of the paper “Cross-Border Mergers and Acquisitions as a Challenge for Sustainable Business” submitted to the Sustainability.

I appreciated the authors' effort in rewriting important parts of the paper (introduction, literature review with two hypotheses, some parts of the methodology and conclusion section). However, the work still needs an effort with reference to methodology and conclusions.

In particular,

The methodology should be supported by previous studies that have jointly analyzed different countries (EU and USA). The main criticism is connected to the fact that your results do not allow a generalization since the countries are very heterogeneous. Please consider it. Also specify the sectors of the companies in a final table.

In the paragraph 3 the phrase “The size of these companies was …… 100 mil” is not clear.

I suggest that you broaden your conclusions by entering the limits of your research and more theoretical implications also in term of sustainability.

Finally, please attach the questionnaire sent to the companies.

Best regards

Author Response

Dear Reviewer,

thank you very much for the comments. We have followed each of them thouroughly and incorporated the revisions into our manuscript - all the amendments are highlighted in yellow.

Round  3

Reviewer 2 Report

Dear Authors,

I am glad to inform you that I have appreciated the paper in the new version. I'm glad to announce your recommendation for the acceptance on the Journal.

Best regards,

This manuscript is a resubmission of an earlier submission. The following is a list of the peer review reports and author responses from that submission.